# CoNO: Complex Neural Operator for Continuous Dynamical Systems

## Abstract

Neural operators extend data-driven models to map between infinite-dimensional functional spaces. These models have successfully solved continuous dynamical systems represented by differential equations, *viz* weather forecasting, fluid flow, or solid mechanics. However, the existing operators still rely on real space, thereby losing rich representations potentially captured in the complex space by functional transforms. In this paper, we introduce a Complex Neural Operator (CoNO), that parameterizes the integral kernel in the complex fractional Fourier domain. Additionally, the model employing a complex-valued neural network along with aliasing-free activation functions preserves the complex values and complex algebraic properties, thereby enabling improved representation, robustness to noise, and generalization. We show that the model effectively captures the underlying partial differential equation with a single complex fractional Fourier transform. We perform an extensive empirical evaluation of CoNO on several datasets and additional tasks such as zero-shot super-resolution, evaluation of out-of-distribution data, data efficiency, and robustness to noise. CoNO exhibits comparable or superior performance to all the state-of-the-art models in these tasks. Altogether, CoNO presents a robust and superior model for modeling continuous dynamical systems, providing a fillip to scientific machine learning. Our code implementation is available at https://anonymous.4open.science/r/anonymous-cono.

## 1 Introduction

Continuous dynamical systems span various scientific and engineering fields, such as physical simulations, molecular biology, climatic modeling, and fluid dynamics, among others Debnath & Debnath (2005). These systems are mathematically represented using PDEs, which are numerically solved to obtain the system's time evolution. The resolution of PDEs necessitates the identification of an optimal solution operator, which maps from functional spaces encompassing initial conditions and coefficients. Achieving this mapping entails discretization procedures for the data capture. Traditionally, numerical methods, such as finite element and spectral methods, have been employed to approximate the solution operator for PDEs. However, these approaches often incur high computational costs and exhibit limited adaptability to arbitrary resolutions and geometries (Sewell, 2012; Ŝolín, 2005).

Recently, neural operators have shown promise in solving these PDEs in a data-driven fashion (Kovachki et al., 2021). Neural operators extend the neural network to map between infinite dimensional functional space and are a universal approximation of the operator (Kovachki et al., 2021). Operator learning was first proposed by Lu et al. (2021), namely, DeepOnet, which theoretically established the universal approximation of operators. DeepONet consists of a branch net and trunk net, where the branch learns the input function operator, and the trunk learns the function space onto which it is projected. Another widely used architecture, Fourier Neural Operator (FNO), a frequency domain method (FDM), was proposed by (Li et al., 2020), which consists of uplifting Fourier kernel-based integral using fast Fourier transform and projection blocks. Following this, several frequency transformation-based kernel integral neural operators have been proposed. For instance, Fanaskov & Oseledets (2022) introduced spectral methods such as Chebyshev and Fourier series to avoid aliasing error and opaque output in FNO, Tripura & Chakraborty (2022) blends integral kernels with wavelet transformation, which uses time-frequency wavelet localization.

Despite several successes of operator learning for solving PDEs, Bartolucci et al. (2023) showed that several problems persist and must be addressed. These include aliasing errors, generalization, robustness to noise, and structure-preserving equivalent operator (Bartolucci et al., 2023; Fanaskov & Oseledets, 2022). Operators must respect the continuous-discrete equivalence while learning the underlying operation, not just a discretized version. This is challenging since the data used for training the operator is provided as a discretized version of the continuous field. As at any finite resolution, the possible mismatch between the continuous-discrete version should not introduce any lead-in error, i.e., this should not lead to any aliasing error (Michaeli et al., 2023). Several operators, such as FNO, suffer from continuous-discrete equivalence, i.e., aliasing error, introduced by pointwise activation.

In the realm of data-driven methods, introducing a learnable order for the fractional discrete Fourier transform (Candan et al., 2000), which facilitates the seamless integration of features between the time and frequency domains, has been an open research problem. However, this concept becomes less clear when applied to the study of continuous systems, representing a broader, more generalized form of the Fourier transform. Recent research by Raonić et al. (2023) has significantly addressed the challenge of aliasing errors within the operator framework. Their approach involves the utilization of convolution operators explicitly parameterized in the physical space, diverging from traditional frequency domain methods. Moreover, they have harnessed UNET-based architectures (Ronneberger et al., 2015) to enhance the architecture's efficiency and memory utilization. It is also worth noting that the study conducted by Shafiq & Gu (2022) emphasizes the critical role of overparametrization in achieving superior generalization and optimization performance.

Another aspect that has received less attention is the complex representation of FDMs. The traditional FDMs in operator learning, including FNO, do not perform non-linear transformations on the complex representations of the Fourier transform. Thus, these models do not exploit the rich representations of complex numbers. The allure of complex number representations has grown considerably due to their ability to capture richer information through phase information (Nitta, 2009) in complex neural networks (Hirose, 2012). They exhibit advantages such as faster convergence (Danihelka et al., 2016; Arjovsky et al., 2016) and improved generalization (Hirose & Yoshida, 2012). While Trabelsi et al. (2017) highlighted the benefits of combining complex neural networks with their real representation counterparts, their application in the Operator learning framework remains largely unexplored.

**Our Contributions.** To address these challenges, we present an operator that utilizes Complex neural networks based on frequency domain representations in this work. Specifically, we introduce the Complex Neural Operator (CoNO), a novel deep learning operator designed to establish mappings between infinite-dimensional functional spaces. Table 1 compares several features of CoNO with other existing operators. Our contributions are outlined as follows:

1. **Complex Neural Operator:** CoNO represents the first instance of a Complex Neural Operator that performs operator learning employing a complex neural network.

2. **Fractional Kernel Integral:** CoNO parameterizes the integral kernel within the complex fractional Fourier transform using a single transformation operation with learnable fractional parameters, thereby reducing the number of transformations in comparison to previous operators.

3. **Data efficiency and Robustness to noise:** CoNO demonstrates high generalization even with minimal samples, data instances, and training epochs. Specifically, CoNO gives the same performance as FNO with 1/4 size of the training data. Further, CoNO exhibits improved robustness to noise in the training or testing dataset compared to SOTA operators.

## 2 PRELIMINARIES

### 2.1 OPERATOR LEARNING FRAMEWORK

**Problem Setting:** We have followed and adopted the notations in Li et al. (2020). Let us denote a bounded open set as $D \subset \mathbb{R}^d$, with $A = A(D; \mathbb{R}^{d_a})$ and $U = U(D; \mathbb{R}^{d_u})$ as separable Banach spaces of functions, representing elements in $\mathbb{R}^{d_a}$ and $\mathbb{R}^{d_u}$, respectively. Consider $G^\dagger : A \to U$ to be a nonlinear mapping, arising from the solution operator for a parametric partial differential equation

| Operators | FDM | Alias Free | Learnable Order | Downscaling | Complex Rep |
|-----------|-----|------------|-----------------|-------------|-------------|
| DeepONet | ✗ | ✗ | ✗ | ✗ | ✗ |
| FNO | ✓ | ✗ | ✗ | ✗ | ✗ |
| WNO | ✓ | ✗ | ✗ | ✗ | ✗ |
| SNO | ✓ | ✓ | ✗ | ✗ | ✗ |
| CoNO (Ours) | ✓ | ✓ | ✓ | ✓ | ✓ |

Table 1: Comparison of the features of different neural operators with CoNO.

(PDE). It is assumed that there is access to independent and identically distributed observations $(a_j, u_j)_{j=1}^N$, where $a_j \sim \mu$, drawn from the underlying probability measure $\mu$ supported on $A$, and $u_j = G^\dagger(a_j)$.

The objective of operator learning is to construct an approximation for $G^\dagger$ via a parametric mapping $G : A \times \Theta \to U$, or equivalently, $G_\theta : A \to U, \theta \in \Theta$, within a finite-dimensional parameter space $\Theta$. The aim is to select $\theta^\dagger \in \Theta$ such that $G(\cdot, \theta^\dagger) = G_\theta^\dagger \approx G^\dagger$. This framework facilitates learning in infinite dimensional spaces as the solution to the optimization problem in Equation 1 constructed using a with a loss function $L : U \times U \to \mathbb{R}$.

$$\min_{\theta \in \Theta} \mathbb{E}_{a \sim \mu} \left[ L(G(a, \theta), G^\dagger(a)) \right], \tag{1}$$

In neural operator learning frameworks, the above optimization problem is solved using a data-driven empirical approximation of the loss function akin to the regular supervised learning approach using train-test observations. Usually, $G_\theta$ is parameterized using deep neural networks.

## 2.2 Complex Neural Networks

In our proposal, we use complex neural networks for approximating $G_\theta$. Here, each neuron will separately output a real and an imaginary part. As demonstrated by Nitta (2002) Nitta (2009), complex neural networks often outperform their real-valued counterparts on function approximation tasks. Notably, since they have both real and imaginary parts, they can facilitate learning of mutually orthogonal decision boundaries in the real and imaginary domains, thereby enhancing generalization capabilities. Furthermore, it was observed that critical points in complex neural networks predominantly manifest as saddle points rather than local minima, in contrast to real-valued neural networks Nitta (2002; 2009). It is noted that stochastic gradient-based optimization algorithms such as SGD can largely avoid saddle points but not local minima (Lee et al., 2016; Jin et al., 2017). Additionally, complex neural networks exhibit improved training efficiency and enhanced generalization compared to standard CNNs Ko et al. (2022).

Note that the activation functions employed in complex neural networks should also respect the complex operations. In our method, we incorporate Complex GeLU (CGeLU) activation functions, which apply independent GeLU (Hendrycks & Gimpel, 2016) (Lee, 2023) functions to a neuron's real and imaginary components, respectively. Formally, this can be expressed as:

$$\text{CGeLU}(z) = \text{GeLU}(\text{Re}(z)) + i\text{GeLU}(\text{Im}(z)). \tag{2}$$

The CGeLU activation function satisfies the Cauchy-Riemann equations when the real and imaginary parts are strictly positive or negative.

## 2.3 Fractional Fourier transform

In the Operator Learning framework, Operators are defined via architectures comprising functional compositions of integral transforms and nonlinear activation functions in the operator learning framework. For instance, while in DeepONet Lu et al. (2021), integral transforms happen in the physical domain, FNO Li et al. (2020) incorporates integral transforms in the frequency domain. In our architecture, we propose to use the Discrete Fractional Fourier Transform (FrFT) with learnable order as the integral transform. This enables learning anywhere 'in between' the physical and the

frequency domains. This transform can be of great interest in deep learning due to its ability to capture different types of frequency content and directional features present in data.

Fractional Fourier Transform (FrFT) is a mathematical operation that generalizes the classical Fourier transform by introducing a parameter that controls the transform's degree of rotation (Ozaktas et al., 1996). Formally, the FrFT of a function $f(x)$ with respect to the fractional order $\alpha$ is defined as:

$$\mathcal{F}_\alpha[f](u) = \sqrt{1 - i\cot(\alpha)}e^{i\pi\cot(\alpha)u^2} \int e^{-2\pi i\left(\csc(\alpha)ut - \frac{\cot(\alpha)}{2}x^2\right)} f(t)dt \ . \tag{3}$$

In the above Equation 3, $\alpha$ is the fractional order, $u$ is the transformed variable, and $\text{sgn}(t)$ is the signum function applied to the variable $t$.

### 2.4 MITIGATION OF ALIASING

The operator learning framework necessitates approximation through non-linear operations, including non-linear pointwise activations, which may introduce arbitrarily high-frequency components into the output signal. The emergence of nonlinearity-induced aliasing can precipitate the symmetry distortion inherent in the physical signal, consequently leading to adverse effects. Moreover, translational invariance, desired in a neural operator, is susceptible to degradation due to aliasing (Karras et al., 2021; Bartolucci et al., 2023).

We employ a two-step process to mitigate aliasing errors within the operator learning paradigm for continuous equivariance. First, before applying any activation function, we upsample the input function at a rate exceeding its frequency bandwidth. Subsequently, we apply a non-linear operation to the upsampled signal, then apply a sinc-based filter (Yaroslavsky, 2002) followed by downsampling. The sinc-based low-pass filter effectively attenuates higher frequency components in the output signal, thus averting aliasing artifacts and preserving the complex domain information. This approach minimizes aliasing in the operator learning framework, as demonstrated empirically later (Sec. 4.6), maintaining the fidelity and integrity of the physical signal.

## 3 COMPLEX NEURAL OPERATOR (CONO)

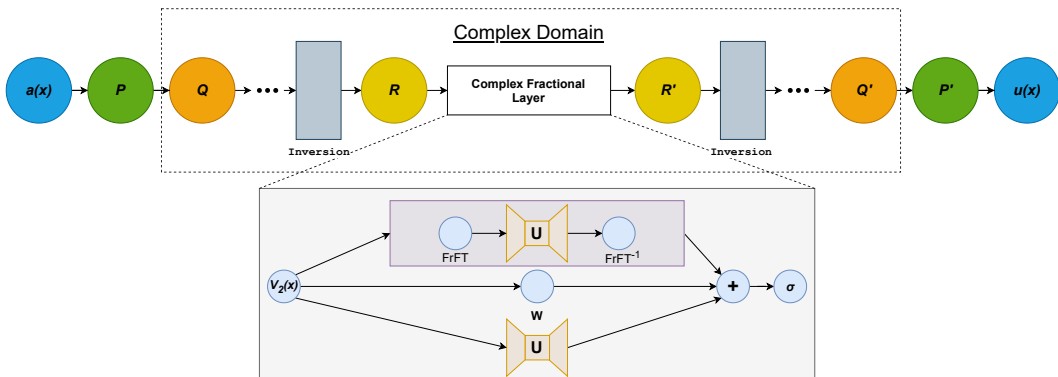

Figure 1: **The full architecture of CoNO.** (1) Input function $a(x)$ is projected into higher dimension through $P$ operation. (2) $P$ is passed through the $R$ operation, which converts the embedding from a real to a complex domain. (3) Lift operation is applied to $Q$ in the complex domain to obtain $v_2(x)$ (4) $v_2(x)$ is passed through a complex fractional integral operator with learnable order parameters where $U$ denotes complex UNET. (5) Then, projection operation $Q'$ is applied to the output. (6) Then it is passed through operation $R'$, which converts the output from the complex to the real domain. (7) Lastly, the operation $P'$ maps to output function $u$. Inversion denotes the discretization inversion operation on which the complex integral transform is trained during super-resolution.

In this subsection, we introduce our proposed architecture, Complex Neural Operator CoNO. Our goal is to construct the operator in a structure-preserving manner where we band-limit a function over a given spectrum (Vetterli et al.), thus preserving complex continuous-discrete equivalence such that Shannon-Whittaker-Kotel'nikov theorem is obeyed for all continuous operations (Unser, 2000).

Let $a : \mathcal{D}_A \to \mathbb{R}^{d_A}$ denote the input function. We initiate the procedure by applying a point-wise operator $P$ to $a$ and obtaining $v_0 : \mathcal{D}_A \to \mathbb{R}^{d_0}$. The point-wise operator $P$ is parameterized as $P_\theta : \mathbb{R}^{d_A} \to \mathbb{R}^{d_0}$, which operates as $v_0(x) = P_\theta(a(x))$ for $x \in \mathcal{D}_0$, where $\mathcal{D}_0 = \mathcal{D}_A$. Typically, $P_\theta$ is realized as a deep neural network. In this paper, we set $d_0 \gg d_A$, designating $P$ as a lifting operator.

Subsequently, we apply the operator $Q$ to $v_0(x)$. This operator is realized as a Complex Convolutional Neural Network (CCNN) with a residual connection, as depicted in Figure 1. This facilitates discretized inversion, thereby maintaining continuous-discrete equivalence through functional interpolation in the complex domain. We compute $v_1 : \mathcal{D}_{v_0} \to \mathbb{R}^{d_1}$ using $v_1(x) = Q_\theta(v_0(x))$ for $x \in \mathcal{D}_{v_0}$, where $Q_\theta : \mathbb{R}^{d_{v_0}} \to \mathbb{R}^{d_{v_1}}$.

Subsequent to this, we employ a complex point-wise operator $R$ on $v_1$ to obatain $v_2 : \mathcal{D}_1 \to \mathbb{R}^{d_2}$. The point-wise operator $R$ is parameterized as $R_\theta : \mathbb{R}^{d_1} \to \mathbb{R}^{d_2}$, operating as $v_2(x) = R_\theta(v_1(x))$ for $x \in \mathcal{D}_1$, implemented as a complex deep neural network. In this work, we set $d_2 = d_1$.

Following this, we employ a complex fractional nonlinear operator on $v_2(x)$, to obtain $v_3(x)$, which involves a complex UNET-shaped operator with a $3 \times 3$ kernel size. During upsampling, zero padding is applied to the signal and convolved with a sinc-based low-pass filter. Downsampling, on the other hand, involves removing the signal components outside the spectrum. This process is implemented using filter-based convolution, accompanied by removing the signal samples at the indices corresponding to padding done during upsampling.

Specifically, $v_3 : \mathcal{D}_{v_2} \to \mathbb{R}^{d_3}$ is calculated as $v_3(x) = Wv_2(x) + K(\alpha; \phi)v_2(x) + K'(\phi)v_2(x)$, where $K(\alpha; \phi)$ and $K'(\phi)$ are kernel integral transformation and convolutional operators, respectively, parameterized by complex neural networks. Here, $\alpha$ represents the fractional complex Fourier transform with a learnable order parameter, and $W$ denotes pointwise complex convolution. In the complex UNET encoding stage, the input function is mapped to vector-valued functions characterized by increasingly contracted domains and higher-dimensional co-domains. Specifically, for each $i$, we have $\mu(D_i) \geq \mu(D_{i+1})$ and $d_{v_{i+1}} \geq d_{v_i}$. For this to be feasible, in this study, without loss of generality, we adopt the Lebesgue measure $\mu$ for $\mu_i$'s.

Further, we apply the $R'$ operation to $v_3$, leading to the computation of $v_4 : \mathcal{D}_3 \to \mathbb{R}^{d_4}$. The point-wise operator $R'$ is parameterized as $R'_\theta : \mathbb{R}^{d_3} \to \mathbb{R}^{d_4}$, acting as $v_4(x) = R_\theta(v_3(x))$ for $x \in \mathcal{D}_3$. Typically, $R'_\theta$ is realized as a complex deep neural network. In this paper, we set $d_3 = d_4$.

Lastly, we employ a Complex CNN with a residual connection to transition from the complex domain to the real domain. This results in the computation of $v_5 : \mathcal{D}_{v_4} \to \mathbb{R}^{d_5}$, where $v_5(x) = Q'_\theta(v_4(x))$ for $x \in \mathcal{D}_{v_4}$, and $Q'_\theta : \mathbb{R}^{d_{v_4}} \to \mathbb{R}^{d_{v_5}}$. Finally, we utilize the $P'$ projection operator to map back to the solution domain $u(x)$, resulting in $u : \mathcal{D}_{v_5} \to \mathbb{R}^{d_u}$ with $u(x) = P'_\theta(v_5(x))$ for $x \in \mathcal{D}_{v_5}$. The entire architectural details of CoNO are depicted in Fig. 1.

## 4 NUMERICAL EXPERIMENTS AND RESULTS

This section presents a comprehensive empirical analysis of CoNO compared to various neural operator baselines, mainly including FDMs, DeepONet(Lu et al., 2021), FNO(Li et al., 2020), Wavelet NO (WNO) (Tripura & Chakraborty, 2022), and Spectral NO (SNO)(Fanaskov & Oseledets, 2022), on standard datasets. We ensure a diverse selection of partial differential equations (PDEs) taken from Takamoto et al. (2022), encompassing both time-dependent and time-independent problems, to account for the intrinsic computational complexity of the tasks. Further, we evaluate CoNO on several tasks, such as performance on out-of-distribution datasets, data efficiency, and robustness to noise. Finally, we perform ablation studies to understand the contribution of several architectural features in CoNO, such as complex neural network, fractional Fourier transform, aliasing-free activation function, and bias towards its final performance. All the experiments are conducted on a Linux machine running Ubuntu 20.04.3 LTS on an Intel(R) Core(TM) i9-10900X processor and NVIDIA RTX A6000 GPUs with 48 GB RAM. All the datasets and codes used in this work are available at CoNO.

### 4.1 Datasets Description and Baseline Experiments

**Setting:** In our experiment, we have leveraged a diverse set of partial differential equations (PDEs), encompassing time-independent models, including Burgers and Darcy Flow, and time-dependent models, such as Navier-Stokes, shallow water, and diffusion equations from Takamoto et al. (2022). This wide-ranging assortment of PDEs has been carefully chosen to facilitate a comprehensive evaluation of the efficacy of our proposed methods. The relative L2 error is presented in Section 4.1.1. The descriptions of datasets used in the present work are as follows.

**Burger's Dataset**: The flow of a viscous fluid in one dimension is modeled by a nonlinear PDE as

$$\frac{\partial u}{\partial t}(x,t) + \frac{\partial}{\partial x}\left(\frac{u^2(x,t)}{2}\right) = \nu \frac{\partial^2 u}{\partial x^2}(x,t), \quad x \in (0,1), , t \in (0,1] \tag{4}$$

This equation, referred to as the 1D Burger's equation, is numerically solved to generate the dataset. This dataset presents the time evolution for one timestep of the equation for a given initial condition. Thus, we aim to learn an operator that maps the initial condition to the next time step. The dataset consists of 2048 training and testing data.

**Darcy Flow Dataset**: Another widely used dataset, Darcy's equation, represents the flow through porous media. 2D Darcy flow over a unit square is given by

$$\nabla \cdot (a(x)\nabla u(x)) = f(x), \quad x \in (0,1)^2, \tag{5}$$

$$u(x) = 0, \quad x \in \partial(0,1)^2. \tag{6}$$

where $a(x)$ is the viscosity, $f(x)$ is the forcing term, and $u(x)$ is the solution. This dataset employs a constant value of forcing term $F(x) = \beta$. Further, Equation 5 is modified in the form of a temporal evolution as

$$\partial_t u(x,t) - \nabla \cdot (a(x)\nabla u(x,t)) = f(x), \quad x \in (0,1)^2, \tag{7}$$

Thus, the goal on this dataset is to learn the operator that maps the diffusion coefficient to the solution. The dataset consists of 10,000 training and testing data.

**Navier Stokes Dataset:** 2D Navier-Stokes equation describes the flow of a viscous, incompressible fluid in vorticity form on the unit torus as

$$\partial_t w(x,t) + u(x,t) \cdot \nabla w(x,t) = \nu \Delta w(x,t) + f(x), \quad x \in (0,1)^2, t \in (0,T] \tag{8}$$

$$\nabla \cdot u(x,t) = 0, \quad x \in (0,1)^2, t \in [0,T] \tag{9}$$

$$w(x,0) = w_0(x), \quad x \in (0,1)^2 \tag{10}$$

where, $u$ represents the velocity field, $w = \nabla \times u$ is the vorticity, $w_0$ is the initial vorticity, $\nu$ is the viscosity coefficient, and $f$ is the forcing function. The goal on this dataset is to learn the operator $G^\dagger$, mapping the vorticity up to time 10 to the vorticity up to some later time $T > 10$. The training and test data in this dataset comprises 5,000 samples with 50 timestamps.

**Shallow Water Dataset:** Compressible Navier-Stokes equations, that model free-surface flow in the form of hyperbolic PDEs, can be used to model shallow water in 2D as

$$\partial_t h + \nabla \cdot (hu) = 0, \tag{11}$$

$$\partial_t hu + \nabla \cdot \left(u^2 h + \frac{1}{2}grh^2\right) = -grh\nabla b, \tag{12}$$

where, $u$ and $v$ denote velocities in the horizontal and vertical directions, $h$ represents water depth, and $b$ characterizes spatially varying bathymetry. The quantity $hu$ corresponds to directional momentum, and $g$ denotes gravitational acceleration. Similar to the previous dataset, the goal on this dataset is to learn the operator $G^\dagger$, which maps velocity up to time 10 to the vorticity up to some later time $T > 10$. The training and test data comprises 1,000 samples with 101 timestamps.

**Diffusion Reaction Equation Dataset:** Consider a 2D domain characterized by two non-linearly coupled variables, namely, the activator $u = u(t,x,y)$ and the inhibitor $v = v(t,x,y)$. Assuming that their dynamics are governed by the equations given below

$$\partial_t u = D_u \partial_{xx} u + D_u \partial_{yy} u + R_u, \tag{13}$$

$$\partial_t v = D_v \partial_{xx} v + D_v \partial_{yy} v + R_v, \tag{14}$$

the goal is to learn the operator responsible for mapping the activator's and inhibitor's initial conditions to their respective states later $T > 0$. The training and test data comprises 1,000 samples with

101 timestamps.

**Metric:** The metric used for experimentation is the mean relative L2 error

$$L = \frac{1}{n} \sum_{j=1}^{n} \frac{\|\hat{u}(j) - u(j)\|_2^2}{\|u(j)\|_2^2} \tag{15}$$

where $n$ is the size of the training data, $u(j)$ represents the $j$-th ground truth sample of the training data, $\hat{u}(j)$ represents the $j$-th sample prediction.

**Training Details and Baselines**: We adhere to standard experimental practices by splitting the dataset into training, testing, and validation sets in ratios of 0.8, 0.1, and 0.1. We employ an ensemble training approach to maintain a level playing field for each operator. This involves specifying a hyperparameter range and randomly selecting a subset of hyperparameters. For the experiments, we use Adam optimizer (Kingma & Ba, 2014). We conduct model training for each optimal hyperparameter configuration using random seeds and data splits. And the weight of the best-performing model on the evaluation. Each experiment is repeated three times, and the mean of relative L2 loss is reported.

### 4.1.1 COMPARISON WITH BASELINES

First, we evaluate the performance CONO in comparison to baselines on the five PDE datasets considered. Table 2 shows that CONO outperforms the baselines on all the datasets except Burgers. We observe among the baselines, FNO outperforms all other models consistently. In Burgers, the performance of FNO and CONO are comparable with FNO being slightly better. However, it is worth noting that the Burgers dataset corresponds to a 1D equation, which is lower dimensional than all other datasets. These results suggest that CONO outperforms FNO in more complex and higher dimensional PDEs, while FNO and CONO may have similar performance in simpler, low dimensional PDEs.

| Datasets | DeepONet | FNO | WNO | SNO | CONO(Ours) |
|----------|----------|-----|-----|-----|------------|
| **Burgers** | $0.027_{\pm 0.002}$ | $0.021_{\pm 0.003}$ | $0.032_{\pm 0.001}$ | $0.23_{\pm 0.01}$ | $0.022_{\pm 0.002}$ |
| **Darcy** | $0.028_{\pm 0.001}$ | $0.024_{\pm 0.003}$ | $0.054_{\pm 0.004}$ | $0.61_{\pm 0.02}$ | $0.021_{\pm 0.003}$ |
| **Navier Stokes** | $0.65_{\pm 0.02}$ | $0.41_{\pm 0.02}$ | $0.73_{\pm 0.01}$ | $8.4_{\pm 0.2}$ | $0.36_{\pm 0.01}$ |
| **Shallow Water** | $0.0064_{\pm 0.0003}$ | $0.00049_{\pm 0.00004}$ | $0.0074_{\pm 0.0005}$ | $0.032_{\pm 0.001}$ | $0.00047_{\pm 0.00003}$ |
| **Diffusion** | $0.92_{\pm 0.01}$ | $0.91_{\pm 0.02}$ | $0.95_{\pm 0.02}$ | $7.3_{\pm 0.1}$ | $0.89_{\pm 0.01}$ |

Table 2: Relative L2 error of CONO and other baselines for different PDE datasets. The best result is highlighted in blue and the second best in orange.

### 4.2 ZERO SHOT SUPERRESOLUTION

The neural operator exhibits mesh invariance, allowing it to undergo training on lower-resolution data and subsequently be applied to higher-resolution data, thereby achieving zero-shot superresolution. CONO has the capability for zero-shot superresolution, while among the baselines only FNO can perform zero-shot superresolution. To this extent, the neural operators were trained on a $128 \times 128$ resolution and tested on higher and lower resolutions for the Darcy flow dataset. Table 3, presents the relative L2 error of CONO and FNO on superresolution. We note that CONO outperforms FNO significantly on all the resolutions. This analysis concludes that the L2 error of CONO does not grow significantly compared to FNO across varying resolutions.

### 4.3 OUT-OF-DISTRIBUTION GENERALIZATION

In this study, we conducted experiments on the Darcy flow dataset, where during training, we set the force term $f$ to a constant value of $\beta = 1.0$. Subsequently, we evaluated the trained model on various values of $\beta$ to assess its out-of-distribution generalization capabilities, as illustrated in Table 4. Remarkably, our results consistently demonstrate that CONO exhibits superior generalization performance compared to other operators.

| Resolution | DeepONet | FNO | WNO | SNO | CoNO(Ours) |
|---|---|---|---|---|---|
| 85x85 | - | $0.052_{\pm 0.002}$ | - | - | $0.044_{\pm 0.001}$ |
| 128x128 | $0.028_{\pm 0.002}$ | $0.024_{\pm 0.002}$ | $0.054_{\pm 0.002}$ | $0.61_{\pm 0.02}$ | $0.021_{\pm 0.002}$ |
| 256x256 | - | $0.039_{\pm 0.002}$ | - | - | $0.029_{\pm 0.002}$ |
| 512x512 | - | $0.058_{\pm 0.002}$ | - | - | $0.043_{\pm 0.002}$ |

Table 3: Relative L2 error for zero-shot superresolution by CoNO and FNO. Note that all the models are trained on $85 \times 85$ resolution and tested on other resolutions. The best result is highlighted in blue and the second best in orange.

| Beta Coeff | DeepONet | FNO | WNO | SNO | CoNO (Ours) |
|---|---|---|---|---|---|
| 0.01 | $0.80_{\pm 0.01}$ | $0.79_{\pm 0.03}$ | $0.80_{\pm 0.03}$ | $10.80_{\pm 0.2}$ | $0.76_{\pm 0.01}$ |
| 1.0 | $0.028_{\pm 0.001}$ | $0.024_{\pm 0.002}$ | $0.054_{\pm 0.003}$ | $0.61_{\pm 0.02}$ | $0.021_{\pm 0.001}$ |
| 10.0 | $0.068_{\pm 0.001}$ | $0.068_{\pm 0.002}$ | $0.084_{\pm 0.003}$ | $0.54_{\pm 0.02}$ | $0.067_{\pm 0.001}$ |
| 100.0 | $0.075_{\pm 0.001}$ | $0.074_{\pm 0.002}$ | $0.089_{\pm 0.003}$ | $0.53_{\pm 0.02}$ | $0.072_{\pm 0.001}$ |

Table 4: Zero shot out of distribution generalization where we trained on beta coeff 1.0 for Darcy Flow and tested for other beta coeff for different Neural Operators. The best result is highlighted in blue and the second best in orange.

## 4.4 Data Efficiency

Now, we evaluate the data efficiency of CoNO in comparison to other baselines. Specifically, we conducted experiments with various training splitting ratios, ranging from 0.8 to 0.2, to investigate data efficiency. All the experiments are performed on the Darcy flow dataset with a forcing term $\beta = 1.0$. Our findings reveal that CoNO consistently outperforms alternatives across all splitting ratios. Notably, with just 20% of the training data, CoNO performs equivalent to FNO (see Tab. 5).

| Ratio | DeepONet | FNO | WNO | SNO | CoNO(Ours) |
|---|---|---|---|---|---|
| 0.8 | $0.028_{\pm 0.001}$ | $0.024_{\pm 0.002}$ | $0.054_{\pm 0.003}$ | $0.61_{\pm 0.02}$ | $0.021_{\pm 0.001}$ |
| 0.6 | $0.029_{\pm 0.01}$ | $0.025_{\pm 0.001}$ | $0.054_{\pm 0.003}$ | $0.64_{\pm 0.02}$ | $0.021_{\pm 0.001}$ |
| 0.4 | $0.031_{\pm 0.001}$ | $0.025_{\pm 0.002}$ | $0.057_{\pm 0.003}$ | $0.65_{\pm 0.02}$ | $0.022_{\pm 0.001}$ |
| 0.2 | $0.034_{\pm 0.001}$ | $0.028_{\pm 0.002}$ | $0.061_{\pm 0.003}$ | $0.67_{\pm 0.02}$ | $0.024_{\pm 0.001}$ |

Table 5: Data Efficiency for the different ratio of dataset which is used for training for different Neural Operators. The best result is highlighted in blue and the second best in orange.

## 4.5 Robustness to Noise

In this study, we performed experiments introducing different noise levels into the training and testing datasets using Gaussian noise. The noise addition process can be explained as follows: For each input sample denoted as $x(n)$ within the dataset $D$, we modified it by adding Gaussian noise with parameters $\gamma N(0, \sigma_D^2)$. Here, $\sigma_D^2$ represents the variance of the entire dataset, and $\gamma$ indicates the specified noise intensity level.

Further, the models were trained on pristine and noisy data with 1% and 5% noise. These models were then cross-evaluated again on pristine data, and noisy data with 1% and 5% noise, covering all combinations. Our investigation yielded notable results, particularly when evaluating the performance of CoNO in the presence of noise within both the training and testing datasets (see Tab. 6). Remarkably, CoNO exhibits consistent performance irrespective of the presence of noise. Specifically, the noisy training + testing yielded the same result as the pristine dataset confirming the robustness of CoNO to noisy dataset.

| Setting | $\gamma$ % | DeepONet | FNO | WNO | SNO | CoNO (Ours) |
|---|---|---|---|---|---|---|
| | - | $0.028_{\pm 0.001}$ | $0.024_{\pm 0.003}$ | $0.054_{\pm 0.004}$ | $0.61_{\pm 0.02}$ | $0.021_{\pm 0.003}$ |
| Noisy Training | 1% | $0.029_{\pm 0.003}$ | $0.025_{\pm 0.003}$ | $0.054_{\pm 0.004}$ | $0.61_{\pm 0.02}$ | $0.022_{\pm 0.003}$ |
| | 5% | $0.030_{\pm 0.003}$ | $0.026_{\pm 0.003}$ | $0.055_{\pm 0.004}$ | $0.62_{\pm 0.02}$ | $0.023_{\pm 0.003}$ |
| Noisy Testing | 1% | $0.032_{\pm 0.003}$ | $0.028_{\pm 0.003}$ | $0.054_{\pm 0.003}$ | $0.64_{\pm 0.03}$ | $0.022_{\pm 0.003}$ |
| | 5% | $0.033_{\pm 0.003}$ | $0.028_{\pm 0.003}$ | $0.054_{\pm 0.003}$ | $0.65_{\pm 0.03}$ | $0.022_{\pm 0.003}$ |
| Noisy Training + Testing | 1% | $0.030_{\pm 0.003}$ | $0.025_{\pm 0.003}$ | $0.055_{\pm 0.004}$ | $0.62_{\pm 0.02}$ | $0.021_{\pm 0.003}$ |
| | 5% | $0.032_{\pm 0.003}$ | $0.025_{\pm 0.003}$ | $0.055_{\pm 0.004}$ | $0.62_{\pm 0.02}$ | $0.021_{\pm 0.003}$ |

Table 6: Robustness to Gaussian noise during training and testing for different settings for different Neural Operator. The best result is highlighted in blue and the second best in orange.

### 4.6 ABLATION AND COMPARISON

In order to gain insight into the CONO architecture and how different components impact the performance, we perform ablation studies. All ablation experiments were conducted for the Darcy flow dataset with beta coeff 1.0. **Vanilla CONO Architecture:** First, we started with the FNO architecture, where Fourier layers are fused into one layer, and a complex neural network is used in a complex domain. Our findings demonstrate that this configuration consistently performs at a level equivalent to FNO as evident from Table 7. **Fractional Fourier Transform Experiment:** In an alternative experiment, we replaced the CONO transformation layer with a Fourier transform. This modification resulted in a slightly diminished performance compared to the CONO architecture as evident from Table 7. Also, in CONO with the Fractional Fourier transform layer, we found that order along different dimensions after training was 0.98 and 0.97, respectively.

**Analysis of Alias-Free Activation:** We conducted experiments to assess the impact of alias-free activation within our proposed architecture. The results indicate that the absence of alias-free activation leads to degradation in performance of the CONO as depicted in Table 7. **Effect of Bias Removal:** In a separate investigation, we removed the $W$ and $U$ components from the CONO model. This adjustment decreased the model's performance, strongly indicating that bias also plays an important role in the learning dynamics of the system.

| | Relative L2 error |
|---|---|
| Vanilla CoNO | $0.024_{\pm 0.002}$ |
| CoNO - FrFT | $0.024_{\pm 0.001}$ |
| CoNO - Alias Free | $0.022_{\pm 0.003}$ |
| CoNO - Bias | $0.026_{\pm 0.001}$ |
| CoNO | $0.021_{\pm 0.001}$ |

Table 7: Ablation results to study the impact of different components on CONO.

### 5 CONCLUDING INSIGHTS

Altogether, we present a novel operator learning paradigm, namely Complex Neural Operator (CONO), that leverages complex neural networks and the complex fractional Fourier transform as an integral operator, thereby ensuring continuous equivalence. This work demonstrates that the rich representation of complex neural networks can be exploited in the operator learning paradigm to develop robust, data-efficient, and superior neural operators that can learn the function-to-function maps in an improved fashion. CONO outperforms existing operators in terms of performance, zero-shot superresolution, out-of-distribution generalization, and robustness to noise. CONO, thus, paves the way for creating efficient operators for inferring real-time partial differential equations (PDEs).

**Limitations and future work.** Although not demonstrated empirically, the architecture of CONO is capable of effectively downscaling and upscaling the output. Thus, CONO can also be trained with differing input and output resolutions. However, the performance of CONO upon upscaling/downscaling requires further investigation. To further advance our understanding of CONO, it is crucial to delve into the underlying mathematical and algorithmic principles. Specifically, we need to unravel the learning mechanisms within the latent space and provide the theoretical foundation of complex operators. Furthermore, our research presents novel challenges that warrant investigation. These include tackling the initialization procedures for fractional orders, devising streamlined

architectures for complex neural operators, delving into the creation of equivariant complex operators, and elucidating the crucial role played by the fractional Fourier transform in the acquisition of insights into the continuous dynamics of complex systems. These can be pursued as part of future studies.

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

# A    APPENDIX

## A.1    HYPERPARAMETERS FOR TRAINING MODELS

| Model | Optimizer | Scheduler | Learning Rate | Number of Parameters |
|---|---|---|---|---|
| FNO | Adam | StepLR | 0.001 | 1188353 |
| WNO | Adam | StepLR | 0.001 | 5337985 |
| SNO | AdamW | StepLR | 0.001 | 147321 |
| DeepONet | AdamW | StepLR | 0.0001 | 36465408 |
| CoNO | AdamW | StepLR | 0.0001 | 3451201 |

Table 7: Hyperparameters Used for Training the Models
.

## A.2    TRAINING TIME, INFERENCE TIME AND MEMORY USUAGE

| Model | Training Time | Inference Time | Memory Usage (%) |
|---|---|---|---|
| FNO | 18.76 | 0.003 | 95.5 |
| WNO | 116.2 | 0.04 | 94.36 |
| SNO | 10.98 | 0.001 | 90.45 |
| DeepONet | 25.5 | 0.05 | 92.43 |
| CoNO | 219.6 | 0.07 | 96.45 |

Table 8: Training, Inference Time of Model in sec and memory usage of the model while training in percentage.

## A.3    ILLUSTRATION OF TRAINING TASKS

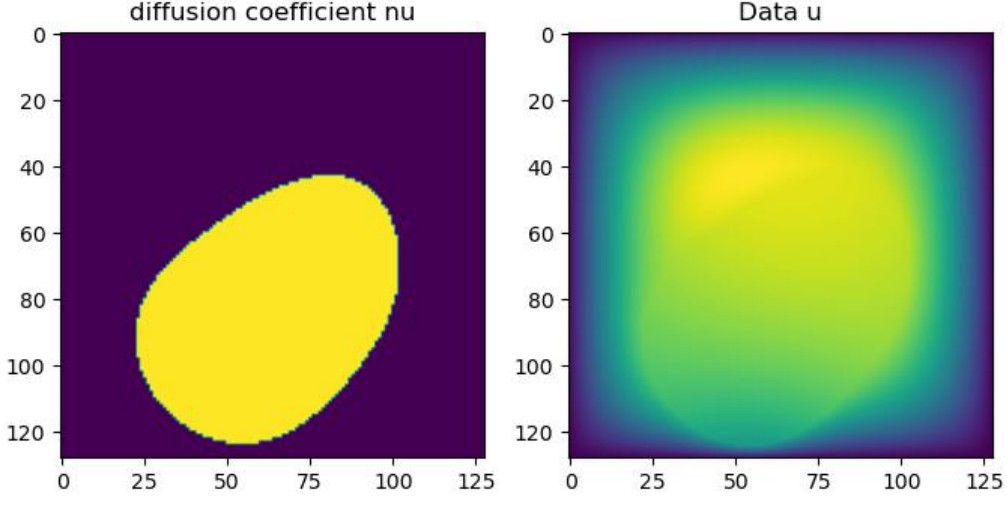

Figure 2:  Illustration of Input and Output dataset of Darcy flow for beta coefficient 1.0

## A.4 CoNO Predictions

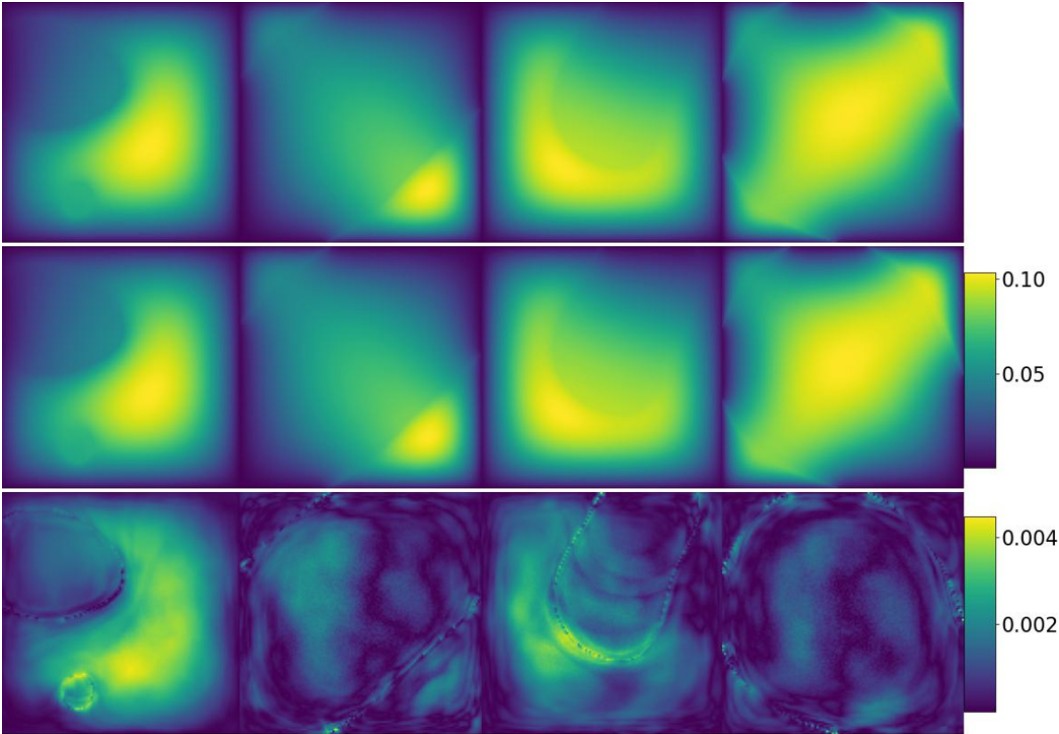

Figure 3: Illustration of Output of the CoNO. (Top) shows the Ground truth. (Middle) shows the models prediction and (Below) shows the error heatmap respectively for Darcy flow dataset.

A.5  ANALYSIS OF LEARNT FRACTIONAL ORDER

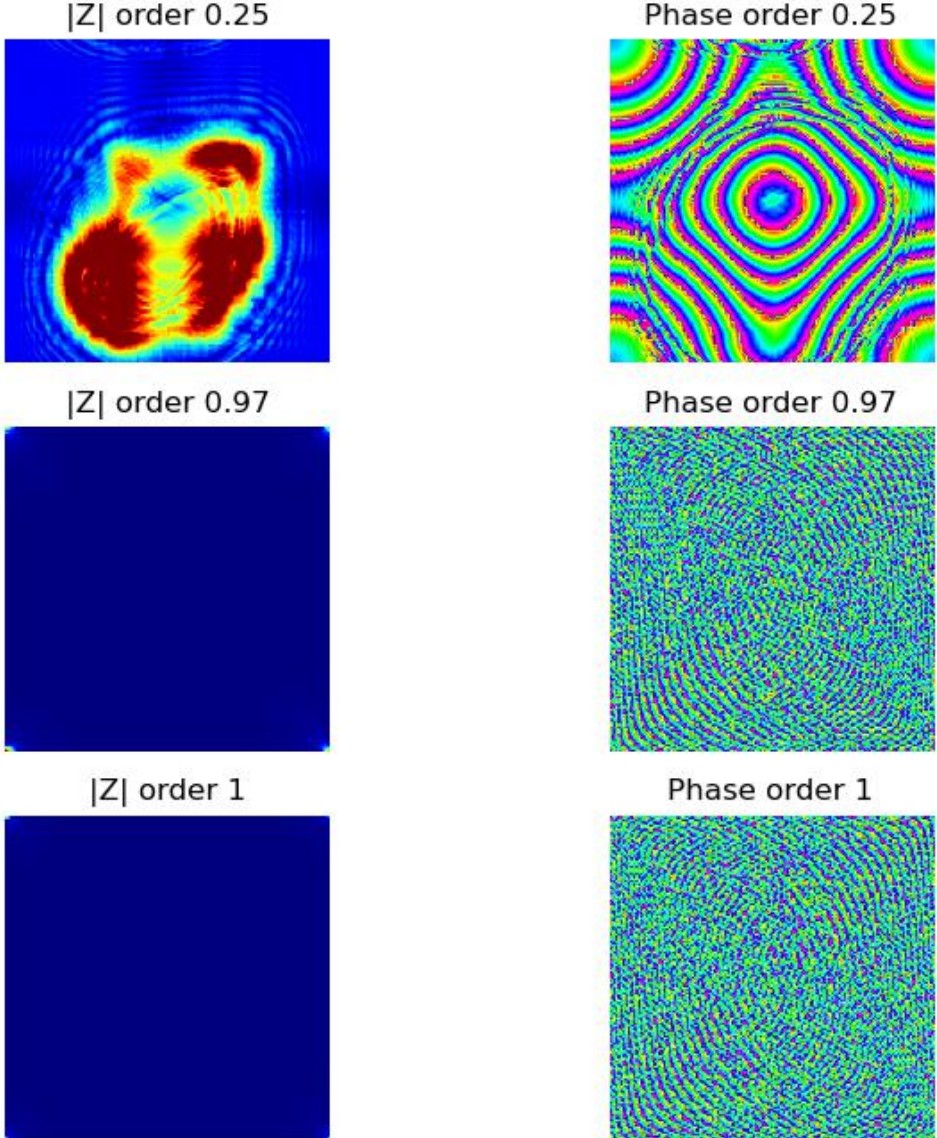

Figure 4: Illustration of Fractional Fourier Transform for different order (left) shows the absolute value of the fractional transform and (right) shows the phase value of the fractional transform for different order for darcy flow dataset.

