# OpenReview forum: "CoNO: Complex Neural Operator for Continuous Dynamical Systems"
_ICLR.cc/2024/Conference — ICLR 2024 Conference Withdrawn Submission_

### Official Review · Reviewer_8MNi · 2023-10-27

**Soundness:** 3 good
**Presentation:** 3 good
**Contribution:** 3 good
**Rating:** 5
**Confidence:** 4

**Summary:**

The CONO is a sophisticated deep learning architecture designed to operate within the complex domain, aiming to capture and represent complex numerical signals effectively. Its architecture is underpinned by several key features:

1. **Complex Domain Operation**: At its core, CONO processes data within the complex plane. By doing so, it taps into the rich information available in both the real and imaginary components of complex data. This enhances its expressive power and capacity for feature extraction.

2. **Point-wise Operators and Transformations**: The model employs several operations and transformations, such as P, Q, and R, which project, convert, and lift data into various domains. Notably, it uses a Complex Convolutional Neural Network (CCNN) for certain transformations and integrates a complex UNET for additional processing.

3. **Fractional Fourier Transforms**: A distinctive feature of the CONO is its utilization of the Discrete Fractional Fourier Transform (FrFT). This allows the model to learn and operate 'in between' the physical and frequency domains, offering a unique perspective and capturing various frequency contents and directional features present in data.

4. **Continuous-Discrete Equivalence**: The model emphasizes maintaining a balance between continuous and discrete operations. This structure-preserving approach ensures that the model remains aligned with foundational principles like the Shannon-Whittaker-Kotel’nikov theorem. This ensures reliable analyses and predictions by the model.

5. **End-to-End Architecture**: From input to output, the CONO model is structured to project, transform, process, and then revert data, ensuring the entire process is smooth and cohesive. The various layers, including complex UNET, CCNN, and point-wise operators, work together in harmony to achieve this.

In summary, the CONO is a robust and versatile model that capitalizes on the richness of complex domain operations, layered transformations, and a structure-preserving approach to effectively handle and analyze complex numerical signals.

**Strengths:**

1. **Complex Domain Operations**: The ability of the CONO model to operate within the complex domain enables it to effectively capture the intricacies of complex numerical signals. This not only augments its expressive power but also enhances feature extraction, making it adept at representing and analyzing complex data.

2. **Structure-preserving Architecture**: The CONO aims to maintain complex continuous-discrete equivalence, ensuring that the Shannon-Whittaker-Kotel’nikov theorem is obeyed for all continuous operations. This kind of structure preservation ensures that the model remains faithful to the underlying physics or principles, making predictions and analyses more reliable.

3. **Comprehensive Framework**: The CONO encompasses a series of intricate operations, transformations, and layers, such as complex UNET, CCNN with a residual connection, and the use of fractional Fourier transforms. This comprehensive framework makes the model versatile and robust, allowing it to handle a wide variety of tasks and challenges, especially in the context of capturing complex numerical signals.

**Weaknesses:**

1. The articulation and expression of the manuscript require further refinement. In several sections, the clarity of the narrative falls short, making it challenging for the reader to grasp the content.

2. The operations of CONO within the complex domain allow it to effectively capture and represent the nuances of complex signals. This enhances its expressive power and improves feature extraction. However, due to the intricacies of the involved operations and transformations, a significant amount of parameter tweaking and experimentation may be necessary to achieve optimal performance. It would be beneficial if the authors could provide detailed settings from their experiments, including memory usage.

3. I'm particularly interested in the experiments related to the NS equation. To my knowledge, the original FNO paper mentioned three distinct viscosity coefficients. However, the authors seem to have chosen 10e-4 without clearly specifying it. This coefficient may not be the most challenging one. I would suggest the authors consider using the more challenging 10e-5 as the viscosity coefficient.

4. I would encourage the authors to incorporate more visualizations to allow readers to gain a more detailed and intuitive understanding of the predicted outcomes.

5. The selection of baselines for comparison appears to be incomplete. To provide a more comprehensive evaluation, I recommend the authors consider adding models like PINN[1] and LSM[2] to the comparisons.

   [1] Raissi, M., Perdikaris, P. & Karniadakis, G.E. (2019). Physics-informed neural networks: A deep learning framework for solving forward and inverse problems involving nonlinear partial differential equations. *Journal of Computational physics*, *378*, pp.686-707.

   [2] Wu, H., Hu, T., Luo, H., Wang, J. & Long, M. (2023). Solving High-Dimensional PDEs with Latent Spectral Models. *arXiv preprint arXiv:2301.12664*.

**Questions:**

see Weaknesses

---

> ### Author Response · Authors · 2023-11-19
> **Response to the comments by Reviewer 8MNi**
>
> **Response:** Thank you for your useful suggestions! We appreciate your positive feedback on the clarity and comprehensibility of our method. We address your concerns point by point below.
>
> 1. We have updated our appendix section for Complex Neural Network and Fractional Fourier Transform for more detail on our manuscript refining approach. Please refer to our updated revision of the manuscript for the same.
>
> 2. We have updated the manuscript revision. Please refer to the revised appendix sections A1, A2, and A3.
>
> 3. Yes, we have chosen the viscosity coefficient 10e-4. Due to time and resource constraints, we conducted the experiments with the NS equation with 10e-5 as the viscosity coefficient. We did it for FNO and CoNO for an input timestamp of 5 and predicted the next 15 timestamps. Here are the results for the same, which demonstrate a slight improvement over FNO.
>
>
> | Dataset | FNO| CoNO |
> | -------- | -------- | -------- |
> | Navier Stokes (1e-5)  | 0.14     | 0.12   |
>
> 4. We have added a few visualizations of the experiments in the appendix. Please refer to the updated revision of the paper in Appendix A3.
>
> 5. We thank the reviewer for the suggestion. We have inherently chosen all frequency-based models for comparison to assess their performance advantages over existing frequency models. In future work, we will add more baseline models and datasets for our comparison.
>
> **Appeal to the reviewer:** If you have any more questions or find some mistakes, please correct us. We sincerely hope you will reconsider and potentially increase the score for our paper.

---

### Official Review · Reviewer_Ubji · 2023-10-29

**Soundness:** 2 fair
**Presentation:** 1 poor
**Contribution:** 2 fair
**Rating:** 3
**Confidence:** 4

**Summary:**

This paper introduces a Complex Neural Operator (CoNO) that parameterizes the integral kernel in the complex fractional Fourier
domain. The authors claim robust and superior modeling of continuous dynamical systems compared to state-of-the-art models in PDE tasks such as Burgers' equation, Navier-Stokes equations, Darcy flow and diffusion.

**Strengths:**

- Fractional Fourier Transform is an interesting concept, yet to be introduced into the deep learning community.

**Weaknesses:**

- There is no information about baseline models and training/inference times. There is a link to the code repo added, but it is impossible to figure out the settings and hyperparamters of the models. Given that the proposed model performs marginally better than FNO models this makes it impossible to judge. Side remark: Pytorch FFTs on complex inputs are much slower than on real inputs (torch.fft vs torch.rfft), thus runtime comparisons would be needed.

- The main formula (Eq 3) is hardly explained. For example in the literature the Fractional Fourier Transform is often defined as $$\mathcal{F}_{\alpha}\[f\](u) = \sqrt{1 - i \cot(\alpha)} e^{i\pi\cot(\alpha)u^2} \int e^{-2\pi i \left( \csc(\alpha) u t - \frac{\cot(\alpha)}{2}x^2\right)} f(t) dt \ .$$  What is the relation of Eq. 3 to the presented formula, and more importantly how can this be implemented? In the code the CoNO model looks very similar to the FNO model, but I assume that the Fourier transform needs to be changed since there is another term which depends on $t$?

- The proposed CoNO model uses a complex UNet part after the fractional transform. It is impossible to guess what brings the claimed performance boost - the fractional transform or the UNet operation in the fractional Fourier domain, which is comparable to pointwise multiplication as done in FNOs? At least comparisons to UNets are therefore inevitable. Especially, since on regular gridded domains UNets / convolutional operators have shown strong performances, see e.g. Raonic et al or Gupta et al.

- Ablation studies are not revealing a lot, they are basically showing the same results as the main table.

- There has been work for example on Clifford Fourier Neural Operators (Brandstetter et al) which includes complex numbers and more complicated algebras. Possibly missing a few others here. Discussions of related work and comparisons against those are missing.

Raonić, B., Molinaro, R., Rohner, T., Mishra, S., & de Bezenac, E. (2023). Convolutional Neural Operators. arXiv preprint arXiv:2302.01178.

Gupta, Jayesh K., and Johannes Brandstetter. "Towards multi-spatiotemporal-scale generalized pde modeling." arXiv preprint arXiv:2209.15616 (2022).

Brandstetter, J., Berg, R. V. D., Welling, M., & Gupta, J. K. (2022). Clifford neural layers for PDE modeling. arXiv preprint arXiv:2209.04934.

**Questions:**

- Why does Equation 3 contain $u$, $x$, and $t$ as variables?
- Can you provide experiments which underline your claim of mitigating aliasing? Or differently put, why is your approach better than FNO? Results don't underline this claim.

---

> ### Author Response · Authors · 2023-11-19
> **Response to the comments by Reviewer Ubji**
>
> **Response:** Thank you for your insightful review. We have updated the manuscript according to the reviews so please refer to the updated version. Here is our response to your concerns.
>
> 1. u is a variable representing a specific point in the frequency domain, x represents a spatial coordinate, and t represents the time domain, respectively.
>
> 2. Please refer to Table 7, which shows the claim of mitigating aliasing. Our approach is better than FNO as it trains with fewer epochs and minimal samples under noisy data samples, as illustrated by our results, unlike FNO, with just 1/4th of the sample size of the dataset.
>
> **Appeal to the reviewer:** If you have any more questions or find some mistakes, please correct us. We sincerely hope you will reconsider and potentially increase the score for our paper.

---

### Official Review · Reviewer_CUfC · 2023-10-29

**Soundness:** 2 fair
**Presentation:** 1 poor
**Contribution:** 2 fair
**Rating:** 3
**Confidence:** 4

**Summary:**

This paper introduces complex neural operator, a new member in the neural operator family, that fully leverages complex representation in the feature space. The kernel integral operation is implemented via fractional Fourier transform. The model is demonstrated to have better simulation accuracy, higher data efficiency and robustness against noise compared with existing neural operators.

**Strengths:**

1. The proposed approach is clean and easy to follow.

2. Complex neural operator permits learnable order and a complete usage of complex representation space.

3. The method seems to perform well in zero shot superresolution and is more robust to noise.

**Weaknesses:**

1. In general the performance improvement of CoNO over FNO seems marginal on most of the datasets. It is not sufficiently convincing to demonstrate the efficacy of the proposed modules. Similar observations can be found in Table 7 for the ablation studies.

2. Lacking theoretical insights on why the model is more robust and performs better on OOD tasks, compared with, e.g., FNO.

3. Presentation can be further improved. For instance, it would be better to index each paragraph in section 3 with a brief summary on what the detailed layer/operation this part is discussing.

4. Lacking some necessary analyses (see Q3, Q4).

Minor: There are indeed multiple typos that do cause difficulty in reading. e.g., Table 4.6 should be Table 7. ``add reference" in page 2. Please fix them.

**Questions:**

Q1. Are there any theoretical insights why the complex model is more robust to noise than FNO?

Q2. It would be better to see more ablations on other datasets since the results in Table 7 is not significant enough especially taking into consideration the stds.

Q3. Can the authors provide visualizations that would give the readers an idea what this task is about and how the model performs qualitatively?

Q4. It would be better to involve analyses on the learned orders, akin to the frequency analyses done in FNO.

---

> ### Author Response · Authors · 2023-11-19
> **Response to the comments by Reviewer CUfC**
>
> **Response:** Thank you for your helpful suggestions! We address your concerns point by point below.
>
> 1. CoNO is more robust to noise as it operates in the complex domain, which offers richer representation and is illustrated by our experiments. For more detailed theoretical insights, please refer to the following literature.
>
> **References**:
> * Chiheb, Trabelsi, O. Bilaniuk, and D. Serdyuk. "Deep, complex networks." International Conference on Learning Representations. 2017.
> * Danihelka, Ivo, et al. "Associative long short-term memory." International conference on machine learning. PMLR, 2016.
> * Wisdom, Scott, et al. "Full-capacity unitary recurrent neural networks." Advances in neural information processing systems 29 (2016).
> * Arjovsky, Martin, Amar Shah, and Yoshua Bengio. "Unitary evolution recurrent neural networks." International conference on machine learning. PMLR, 2016.
>
> 2. We have done more ablation on the proposed method on Navier Stokes for CoNO, which suggests that the problem is that the spectral method is not usually great for modeling the solution as it doesn't help in capturing the nonstationary signal better than FFT, as illustrated by the result below.
>
> | Setting           | Relative L2 error |
> | ----------------- | ----------------- |
> | CoNO              | 0.36              |
> | CoNO - FrFT + FFT | 0.39              |
>
>
>
> 3. We have added a visualization of the proposed method in the appendix. Please refer to Appendix A3 in the revised manuscript version.
>
> 4. We have added a few analyses of the learned fractional order in the appendix, showing the fractional order learned by the model. Please refer to Appendix A4 in the revised version.
>
> **Appeal to the reviewer:** We welcome any questions or corrections you may have and sincerely hope for a reconsideration of our paper's score. Your feedback is highly valuable to us.

---

### Official Review · Reviewer_HK2j · 2023-10-30

**Soundness:** 3 good
**Presentation:** 2 fair
**Contribution:** 2 fair
**Rating:** 5
**Confidence:** 4

**Summary:**

The paper proposes to new paradigm of neural operator learning. The key components are the use of complex numbers in the latent space and the use of fractional Fourier transform as the frequency transform. There are additional components which help in proposing the model as an alias-free method. The performance of the model is evaluated on several standard PDE and it is shown to have comparable/superior performance in several cases, including out-of-distribution, super-resolution, robustness to noise.

**Strengths:**

While there are some notational discrepancies, as pointed out in the section below, I think the authors do a overall good job in sticking to notation and make the method section very clear to understand.

The placement of the proposed method within existing work is adequate.

I think the use of complex numbers along-side fractional Fourier transform is novel and something that is being demonstrated well.

I specifically like section 2.4 where authors describe a simple yet effective anti-aliasing strategy.

**Weaknesses:**

I think the proposed method is novel, and that it can give comparable performance. What is not clear from the presentation both theoretical justification and/or empirical evidence is the benefits that it can have over FNO. Please see questions section for specific.

The writing/overall presentation can be improved.
 - For example, Table 1, caption is not all descriptive, without first introducing what is order of transform, a comparison is being made. This table is not necessary in my opinion.
- After equation (6), $f$ becomes $F$, small things like these
- The description of the several PDE while good, takes up lot of space, which can be otherwise devoted to better explaining and further evaluating the use of complex number

(Minor) There are several typos, some of which I will point out below:
- Reference to Table 7 is broken
- The caption of figure 1 is not coherent with the figure and/or text description of the model. These are important, as readers will get super-confused if these are not in place.
- Beginning of section 2.3, sentence is redundant

**Questions:**

1) What is the comparison in terms of the number of parameters, memory usage, training and/inference time between the proposed method and the baselines, specifically FNO. I think these are showing more/less comparable performance in all cases, and they need more clarity.
    - I feel that because of the use of complex numbers, the model has more capacity but that also makes me think, that the model will be 2 times as more expensive than FNO. In d-dimensions this will be $2^d$ times more expensive. Hence, the claim that authors make that CoNO is better in higher dimensions is questionable in terms of the trade-off.

2) For Navier Stokes and Shallow Water experiment, why do we need the first 10 steps to learn the operator? This seems arbitrary to me and is presented without any justification. In all other cases this is only dependent on the initial condition as should be the case.

3) Comparison to baseline: It is mentioned " randomly selecting a subset of hyperparameters" for models and then one line later "We conduct model training for each optimal hyper-parameter configuration using random seeds and data splits". How is the optimal choice made in each case?

4) As evident in Table7, the use of Fractional Fourier transform doesn't seem to add much of performance boost. What do authors' comment on this choice? Does the learnable order help?

As mentioned previously, I am not able to draw line between CoNO and other Frequency domain method like FNO apart from the use of complex numbers without clear benefit and hence would like clarification and discussion from the authors on these points above.

---

> ### Author Response · Authors · 2023-11-19
> **Response to the comments by Reviewer HK2j**
>
> **Response:** Thank you for the valuable feedback! We greatly appreciate your recognition of the novelty. We have revised all the minor changes as suggested in the latest manuscript. We want to address your concerns point by point as follows.
>
>
> 1. In comparing the model parameters, memory usage, training, and inference time. It is expensive in comparison, as complex pytorch modules are not GPU-optimized. However, during experimentation, we observed that CoNO trains faster than FNO in terms of epochs reaching near equivalent performance within 100 epochs. Please refer to Appendix A1 and A2 in the revised manuscript version.
>
> 2. For Navier Stokes and Shallow Water, we have used the first ten timestamps per the standard practice we followed from the FNO paper. For the Diffusion Reaction, we used one timestamp to verify the robustness of the operator based on initial condition.
>
> **Reference**:
> * Li, Z., Kovachki, N., Azizzadenesheli, K., Liu, B., Bhattacharya, K., Stuart, A., & Anandkumar, A. (2020). Fourier neural operator for parametric partial differential equations. arXiv 2020. arXiv preprint arXiv:2010.08895.
>
> 3. Optimal Choice is made using grid search on a set of defined hyperparameters. " randomly selecting a subset of hyperparameters" means that we have defined a set of hyperparameters for all models and selected the best set of hyperparameters for each model. "We conduct model training for each optimal hyperparameter configuration using random seeds and data splits." Once we have the best set of hyperparameters, we run the model for different random seeds and report the average L2 error for the test set.
>
> 4. As evident from Table 7. We observe that the fractional Fourier transform around the x and y direction is 0.97 and 0.98, respectively, which differs from order 1 in the usual Fourier transform and provides insights that the maximum of the energy spectrum lies near order 1; therefore, not much improvement is seen from Table 7. Yes, the fractional order helps as with fractional order 1 and 1 along both axes we observe slight degradation in performance on Darcy flow, which suggests that the problem is that the spectral method is not usually great for modeling the solution as it helps in capturing the nonstationary signal better than FFT.
>
>
> **Appeal to the reviewer:** If you have any more questions or find some mistakes, please correct us. We sincerely hope you will reconsider and potentially increase the score for our paper.

---

> > ### Comment · Reviewer_HK2j · 2023-11-21
> >
> > Thank you authors for the rebuttal. Unfortunately, I am not convinced by the answers, and hence my score remains the same.

---

### Author Response · Authors · 2023-11-19
**General comments. Applies to all the reviewers**

We thank the reviewers for their careful evaluation and suggestions. Please find a point-by-point response to all the comments raised by the reviewers below. We have also updated the main manuscript and the appendix to address these comments. The changes made in the main manuscript are highlighted in blue color. The major changes made in the manuscript are listed below.

1. **Model Prediction**: We have included the model prediction on the given dataset and ground truth.
2. **Training, Inference, and Memory Usage**: We have included the table highlighting training, inference, and memory usage for each model in the appendix.
3. **Hyperparameters**: We have included the grid search hyperparameters for training each model.
4. **Minor changes**: We have updated some minor typos errors as suggested by each reviewer.